# Impact of Nicotine Replacement and Electronic Nicotine Delivery Systems on Fetal Brain Development

**DOI:** 10.3390/ijerph16245113

**Published:** 2019-12-14

**Authors:** Sebastian Sailer, Giorgia Sebastiani, Vicente Andreu-Férnández, Oscar García-Algar

**Affiliations:** 1Neonatology Unit, Hospital Clinic-Maternitat, ICGON, BCNatal, 08028 Barcelona, Spain; gsebasti@clinic.cat (G.S.); OGARCIAA@clinic.cat (O.G.-A.); 2Grup de Recerca Infancia i Entorn (GRIE), Institut d’investigacions Biomèdiques August Pi i Sunyer (IDIBAPS), 08028 Barcelona, Spain; VIANDREU@clinic.cat; 3Department of Nutrition and Health, Valencian International University, 46002 Valencia, Spain

**Keywords:** E-cigarettes, electronic nicotine delivery systems, nicotine replacement therapy, fetal brain development, pharmacodynamics, pharmacokinetics

## Abstract

Maternal tobacco smoking during pregnancy remains a major public health issue. The neurotoxic properties of nicotine are associated with fetal neurodevelopmental disorders and perinatal morbimortality. Recent research has demonstrated the effects of nicotine toxicity on genetic and epigenetic alterations. Smoking cessation strategies including nicotine replacement therapy (NRT) and electronic nicotine delivery systems (ENDS) show lack of clear evidence of effectiveness and safety in pregnant women. Limited trials using randomized controls concluded that the intermittent use formulation of NRT (gum, sprays, inhaler) in pregnant women is safe because the total dose of nicotine delivered to the fetus is less than continuous-use formulations (transdermal patch). Electronic nicotine delivery systems (ENDS) were hyped as a safer alternative during pregnancy. However, refill liquids of ENDS are suspected to be cytotoxic for the fetus. Animal studies revealed the impact of ENDS on neural stem cells, showing a similar risk of pre- and postnatal neurobiological and neurobehavioral disorders to that associated with the exposure to traditional tobacco smoking during early life. There is currently no clear evidence of impact on fetal brain development, but recent research suggests that the current guidelines should be reconsidered. The safety of NRT and ENDS is increasingly being called into question. In this review, we discuss the special features (pharmacodynamics, pharmacokinetics, and metabolism) of nicotine, NRT, and ENDS during pregnancy and postnatal environmental exposure. Further, we assess their impact on pre- and postnatal neurodevelopment.

## 1. Introduction

Conventional tobacco smoking during pregnancy remains a public health issue with a global prevalence of 1.7%. Differences among countries are considerable, with the highest rates in Ireland (38.4%), Uruguay (29.7%) and Bulgaria (29.4%) and the lowest rates in Tanzania (0.2%), Burundi (0.3%), and Sri Lanka (0.3%). The estimated prevalence for Europe is higher (8.1%) than in America (5.9%) but lower than in Asia (25%) [1,2]. Conventional tobacco cigarette smoke contains more than 4000 chemicals, and 93 of them are classified as potentially harmful [3]. Tobacco smoking during pregnancy is associated with infertility, intrauterine growth restriction, miscarriage, premature birth, and sudden infant death syndrome [4,5]. The U.S. Department of Health and Human Services reported that tobacco smoking during pregnancy causes more than 1000 infant deaths annually [6]. Besides, postnatal secondhand smoking (SHS) is linked to developmental delay and behavioral problems including conduct disturbances and hyperactivity/inattention patterns [7,8]. Recently, SHS was associated with neurosensorial hearing loss due to cochlear affection in patients from 6–12 years even after short-term tobacco exposure [9]. Therefore, tobacco smoking cessation during pregnancy and the prevention of SHS are of utmost importance and should be included in the routine protocols of prenatal controls. First-line cessation strategies include personal counselling as well as cognitive and behavioral therapy, being considered effective and safe [10]. Nicotine delivery systems such as nicotine replacement therapy (NRT), including gums, transdermal patches, nasal sprays, inhaler, and sublingual tablets/lozenges, and electronic nicotine delivery systems (ENDS) deliver nicotine with less toxic chemicals than combusted tobacco leaves [11]. In current guidelines, NRT use during pregnancy remains controverted but is considered a less harmful alternative compared with conventional tobacco smoking if smoking cessation was unsuccessful (Table 1) [12]. ENDS, also known as Electronic Cigarettes, E-cigs, or Vapors, were introduced between 2003 (China) and 2006 (U.S. and Europe), reaching a high popularity [13]. In 2016, the Food and Drug Administration (FDA) regulated ENDS as tobacco products (“deeming regulation”). ENDS consist of a battery, a microprocessor, a heating element, and a fluid-containing reservoir. The reservoir contains the nicotine and a carrier fluid, usually 1,2-propanediol (1,2 PDO). Both substances are heated, vapored, and inhaled by the consumer. ENDS have been steadily evolving. First-generation devices resembled conventional cigarettes and were either disposable or rechargeable. Further, nicotine delivery was insufficient due to poor lung penetration. Second-generation devices (vape pens) were equipped with an e-liquid reservoir and were rechargeable. Later, “Mods” were developed as advanced personal vaporizers, where voltage could be adjusted by the consumer for higher vap temperatures, leading to increased nicotine delivery. Newer-generation devices include JUUL electronic cigarettes, which contain a prefilled e-liquid cartridge and a battery and a temperature regulation device. Further, they use nicotine salts to emulate the nicotine “hit” of conventional cigarettes [14]. ENDS are sometimes used to aid smoking cessation, but their long-term effectiveness and safety remains unclear [15]. ENDS use showed a 7-fold higher risk to start smoking conventional cigarettes among teenagers, but it remains unclear if this increase is caused by the ENDS use itself or by an enhanced willingness to experiment with prohibited substances [16]. Otherwise, NRT and ENDS use during pregnancy remains misunderstood. In 2015, a survey among pregnant women revealed that 43% consider ENDS as less harmful for the fetus than traditional cigarettes [17,18].

The aim of this review is to analyze the current knowledge about NRT and ENDS and their impact on fetal brain development. We include International Guidelines on tobacco smoke cessation and scientific studies performed from the year 2000 until now. Further, we resume insight on pharmacokinetics and dynamics of nicotine, NRT, and ENDS.

## 2. Methods

### 2.1. Literature Search Strategy

We conducted a narrative review with systematic search criteria to identify all studies reporting on NRT and ENDS and their impact on fetal brain development. The search was conducted using the electronic databases PubMed (MeSH) and Embase (Emtree). We used multiple combinations of the following terms: (1) brain *, neuro*, fetal *, fetal *, develop *, cerebr *; AND (2) nicotine replacement therapy *, NRT *, patches *, gum *, inhaler *, spray *, tablets *, lozenges *, electronic nicotine delivery systems *, ENDS *, e-cigarettes, e-cigs, vapors. The search was performed to identify studies published between 1 January 2000 and 15 September 2019, including articles written in English, Spanish, German, and Italian, without geographical restrictions. We included animal studies due to the lack of sufficient studies performed in humans.

### 2.2. Study Selection

Study selection began by screening titles and abstracts for inclusion. Then, full-text articles of all studies (in vitro, animal, human) screened as potentially relevant were considered. Articles were excluded if they (a) lacked information about fetal brain development AND NRT or ENDS; (b) publication date was before 2000. Finally, we also searched for additional literature in the reference lists of the screened articles. Two investigators conducted each step of the study selection. All data were extracted by one investigator and cross-checked by a second investigator. In case of discrepancies with the selected studies, we opted for reconciliation through team discussion.

## 3. Review

### 3.1. Pharmacokinetics and Dynamics of Nicotine, NRT, and ENDS

Since the first human pharmacokinetic studies of nicotine in 1970, a large number of authors have evaluated the use of a wide range of nicotine and tobacco products (including ENDS), using mass spectrometry to analyze blood nicotine concentration levels [19]. Cmax indicates the maximum level of blood nicotine; Tmax shows the time at which maximum nicotine level appears; and AUC (area under the curve) measures the total nicotine exposure in a period of time. Once it reaches the body, nicotine is absorbed through different mucous cell membranes depending on pH. As a weak base (pKa = 7.9), nicotine is not able to cross membranes in acidic environment due to its ionized state. However, at physiological blood pH (7.4), about 30% of nicotine is nonionized and can cross the plasmatic membrane. The most efficient route of absorption is smoking because nicotine enters directly into the circulatory system from the huge alveolar capillary interface and reaches the brain in seconds. After 30 min of exposure, the brain concentration declines due to nicotine distribution to other body tissues. Nicotine gums and chewing tobacco are absorbed through buccal mucosa, where an alkaline pH and the thin epithelium facilitates its absorption, bypassing hepatic metabolism. Swallowed nicotine is absorbed in the small bowel and then metabolized in the liver, generating a significant decrease of bioavailability (30% of total intake) in comparison with previous routes of administration. This fact must be taken into account in oral forms of nicotine replacement therapy [20]. Nicotine distribution after oral, nasal, or transdermal absorption produces, in contrast to inhalation, a gradual increase in nicotine concentrations in the brain [21]. In the liver, nicotine is metabolized to cotinine, mainly by the enzyme cytochrome P450 2A6 (CYP2A6) and to a lesser extent by cytochrome P450 2B6 (CYP2B6) and cytochrome P450 2E1 (CYP2E1) [22]. It is important to highlight that the rate of nicotine metabolism is faster in women than in men, and even faster in pregnant women, increasing the plasma clearance of nicotine up to 60%, and to 140% for cotinine [23]. As a consequence, the implementation of NRTs or ENDs strategies may not be effective to mitigate withdrawal symptoms in pregnant women. Besides, both strategies could need a continued high dosage to reduce this withdrawal. Cotinine can be measured in biological matrices such as blood, saliva, or urine of tobacco consumers. Therefore, cotinine has been largely used as biomarker of nicotine exposure as well as a marker to measure the activity of CYP2A6 in order to report genetic susceptibility to tobacco use, due to the high stability of the molecule (half-life of 16 h) in contrast to nicotine (half-life of 2 h) [24]. In pregnant women, the half-life of cotinine is reduced to 9 h. Otherwise, nicotine shows significant oscillations in blood concentrations from cigarette to cigarette, and it builds up in the body over 8 h. For that reason a total exposure of 24 h must be considered to measure nicotine intake, especially when its effects on nicotinic acetylcholine receptors (nAChRs, which are present very early in the fetal brain) are evaluated [25]. In particular, fetal toxicity of nicotine is due to the fact that this molecule easily crosses the placental barrier, showing a fetal blood concentration (30 min after exposure) up to 15% higher than the maternal concentration, raising to 80% higher in amniotic fluid than in maternal plasma [26,27]. This exposure produces high levels of nicotine in the fetal brain, affecting nAChRs levels as well as the release of neurotransmitters including dopamine, serotonin, acetylcholine, noradrenaline, adrenaline, glutamate, and γ-aminobutyric acid (GABA). Nicotine is also present in breast milk, extending the nicotinic exposure to the postnatal period during breastfeeding [28]. In reference to the pharmacodynamics of nicotine, there are clear differences in its effects at low or high doses. Low doses of nicotine stimulate the central nervous system and increase heart rate and blood pressure. Conversely, high doses depress the central nervous system, producing bradycardia and hypotension [29]. Moreover, low individual tolerance to nicotine produces an increase in heart rate, nausea, and dysphoria, while high tolerance reduces the positive rewards of smoking and increases the withdrawal symptom.

Nicotine replacement therapy (NRT) is available as gum, nasal spray, inhaler, sublingual tablet, and transdermal patch, improving abstinence rates. However, its efficacy is limited by the insufficient dosages, derived from the low efficiency of their routes of absorption. For example, nicotine gum, inhaler, or tablet absorbed through buccal mucosa show plasma nicotine concentrations from one-third to two-thirds of those obtained after smoking a cigarette [30]. Compared with the blood levels of nicotine after smoking a cigarette (10–50 ng/mL), nicotine patches show a concentration from 10 to 20 ng/mL, being from 5 to 15 ng/mL for nicotine gum, sublingual tablet, inhaler, and nasal spray [31,32]. However, these alternative routes of administration produce a much slower decrease in nicotine blood concentrations than cigarettes, showing a total dose of nicotine higher than that of smoking, with the exception of the nasal spray, whose pharmacokinetics are similar to those of cigarettes consumption [30]. ENDS represent the most recent alternative to treat smoking, vaporizing a chemical mixture which contains nicotine (between 0 and 34 mg/mL) to the lungs of the smoker. Nicotine is delivered to the respiratory tract, showing a pharmacokinetic curve similar to that of cigarettes, with oscillations peaks in intervals of 2 h and an accumulative of 24 h. Blood nicotine levels range between 1 and 9 ng/mL and are detectable after 30 s of puffs, decreasing slowly after 30 min until the next series of puffs. However, it must be considered that there exists a clear difference between first and second–third generation of e-cigarettes. The first e-cigarette nicotine pharmacokinetic study was published in 2010, concluding that e-cigarettes showed similar pharmacokinetic patterns of traditional cigarettes, but with significantly lower detectable levels (ten-fold) of nicotine in blood (1.3 ng/mL) [33]. Most recent studies concluded that the first generation of e-cigarettes shows lower Cmax values of nicotine blood concentrations (3.5 ng/mL) than traditional cigarettes (20 ng/mL) in periods of 10 min or less [34]. However, these values are directly influenced by the usage habits, raising similar values to traditional cigarettes in experimented users of e-cigarettes [35]. In contrast with the quick detection of nicotine in blood after smoking a cigarette (from seconds to 5 min), the Cmax takes at least 20 min following initiation of puffing [36]. Second- and third-generation of e-cigarettes show Cmax values similar to those of classical cigarettes. Hajek et al. concluded that a combustible cigarette had a Cmax value of 18 ng/mL, while a newer-generation e-cigarette value was 11.9 ng/mL [37]. Recent works highlight the trend of increased nicotine delivery in NRT and ENDS therapies, allowing users to reach plasma nicotine levels in the same range to tobacco smokers.

### 3.2. Current Guidelines for NRT and ENDS Use during Pregnancy

Scientific evidence concerning NRT and ENDS use during pregnancy is poor due to lack of randomized controlled trials (RCT) in both delivering methods. The Royal Australian College of General Practitioners (RACGP), the National Institute for Health and Care Excellence (NICE), and the Canadian Action Network for the Advancement and the Dissemination and Adoption of Practice-informed Tobacco Treatment (CAN-ADAPTT) guidelines recommend NRT use if previous smoking cessation was unsuccessful. The United States Preventive Services Task Force (USPSTF) and the American College of Obstetricians and Gynecologists (ACOG) do not recommend ENDS due to lack of evidence in pregnant women. Verbiest et al. reviewed current national guidelines for smoking cessation, where 16/21 guidelines considered NRT as a smoking cessation tool in pregnant women. Further intermittent-dosage forms (gum, nasal, and oral sprays) were preferred over continuous-dosage forms (patches). Norway, Scotland, the United States, Japan, and Kyrgyzstan do not recommend NRT during pregnancy [38]. The USPSTF does not recommend pharmacological interventions and ENDS for tobacco cessation in pregnant women due to lack of evidence [39]. The ACOG recommends NRT under close supervision and after careful consideration due to controversial clinical evidence (multiple trials assessing NRT in pregnancy have been stopped by data and safety monitoring committees due to adverse pregnancy effects or failure to demonstrate effectiveness) [40,41]. The available evidence regarding ENDS is even poorer due to the absence of RCTs to determine health effects, smoking cessation capacity as well as the effects on pregnant women and their fetuses [10]. Centers for Disease Control and Prevention (CDC) alerted that the use of ENDS and other nicotine-containing products and even the flavorings may cause brain and lung damage in the fetus [42]. Table 1 shows recommendations on NRT and ENDS use during pregnancy.

### 3.3. Nicotine Replacement Therapy (NRT) 

Nicotine is a molecule with a well-known neurotoxicity that generates harmful effects on fetal/neonatal neurodevelopment. Nicotine has been associated to hyperactive behavior when administrated during the third trimester and during the postnatal period. [47] Currently, the main evidence concerning NRT safety and impact on fetal/neonatal brain development is extrapolated from animal studies (Table 2). Pregnant rats were exposed to either nicotine or saline solutions, whereby the offspring was exposed intra-utero and postnatally (through breastmilk) to nicotine. Synaptic plasticity of offspring was disrupted in utero and in breastfed dams in the NRT group. Further, this trial demonstrated impaired connectivity in the growing brain [28]. In a murine study, Roy et al. administered nicotine during gestation and in the postnatal period (+21 and +30 days). Brain morphology of the dorsal hippocampus and somatosensory cortex showed neuronal maturation and long-lasting alterations in the structure of key brain regions involved in cognition [48]. Further prenatal nicotine exposure elicited persistent suppression of 5HT1A receptors and upregulation of 5HT2 receptors, leading to deficits in the number of neurons, neuronal and synaptic damage, and cognitive dysfunction [49]. Rhesus monkeys exposed pre- and postnatally to environmental tobacco smoke (ETS) showed changes in brain cell development including cell loss (reduced DNA concentration), increased cell size, and replacement of larger neuronal cells with smaller and more numerous glia cells [50]. In a prospective observational study, NRTs were associated to an increased risk for congenital malformations in a non-smokers cohort compared with pregnant women exposed to conventional tobacco smoking [51]. An association between NRT and malformations could not be confirmed in later conducted studies [52,53]. Whereas, no association for increased risk for stillbirth was found [54]. Cooper et al. conducted a two-arm randomized controlled trial (placebo vs. NRT) on 1050 pregnant smokers. NRT showed a “less harmful” impact on behavior, development, and disability in the offspring compared with conventional tobacco smoking after the 2 year follow-up period. Nevertheless, the main objective of the study was the assessment of effectiveness/cost-effectiveness on smoking cessation [55].

Concerning types of NRT administration, intermittent-use formulations including nicotine gum, nicotine spray, and the nicotine inhaler are more preferred than the transdermal patch due to greater total dose of nicotine delivery. The formulation of NRT may affect the level of nicotine in breastmilk. Mothers who use NRT intermittently might minimize the presence of nicotine in breastmilk by prolonging the duration between nicotine administration and breastfeeding [56]. On the mouse model, maternal nicotine exposure showed increased transgenerational metabolic risk including impaired glucose homeostasis, alteration of serum lipids, impaired mitochondrial enzyme activity in skeletal muscle, and elevated blood pressure [57,58]. NRT safety during pregnancy still remains unclear. Currently, there is still lack of studies assessing safety and neurodevelopmental of NRT in humans.

### 3.4. Electronic Nicotine Delivery Systems (ENDS) 

Currently, no studies assessing ENDS’ efficacy and safety for smoking cessation during pregnancy are available [59]. In this review, we analyzed six studies in total, including two studies in vitro and four based on animal models (Table 3). However, we were unable to find studies conducted in humans. Previously, ENDS were linked to impaired placental trophoblast function and diminished alveolar cell proliferation and postnatal lung growth [60,61]. ENDS do not seem to be free from the risk concerning brain development. Nicotine is well known as a hazardous substance to the human body. Therefore “nicotine-free” ENDS are available as a supposedly less harmful alternative. Omaiye et al. assessed the real nicotine concentration of 125 nicotine-free (0 mg of nicotine/mL) ENDS refill fluids from four different countries, detecting counterfeit products containing considerable concentrations of nicotine [62]. Neural stem cell (NSC) mitochondria after ENDS exposure showed more sensitivity to cytotoxic agents in vitro. Moreover, the direct absorption via the olfactory tracks aggravated the oxidative stress and hiperfusion to NSC [63]. In vitro studies concluded that the embryonic stem cells (hESC and mNSC) showed moderate cytotoxity to refill liquids, which indicated an increased risk for embryonic loss or developmental defects during pregnancy [64].

Stem cells are present throughout life and they are more sensitive to stress than differentiated cells. This fact is of crucial importance in organ development and tissue repair. The cellular damage promoted by oxidative stress alters the cellular machinery, causing disease or cell aging in adults. Stem cells of the nervous system are particularly vulnerable to toxicants during development. Therefore, they are excellent for assessing exposure to potential toxicants [65].

In murine studies, ENDS were also linked to behavioral changes including memory and cognition, altered brain development, and neurotransmission deficits. Authors suspected that components other than nicotine affect neurodevelopment, associating ENDS with adverse neurobiological and neurobehavioral outcomes in a similar way to early life conventional cigarettes [66,67]. Nicotine-free aerosol showed a statistically significant higher global DNA methylation pattern, leading to modified gene function by silencing. ENDS were also associated to significant gene expression changes in frontal brain cortex in the murine model [68]. 1,2-PDO is used as a carrier in ENDS and was classified in 2007 as safe by the FDA [69]. This substance is part of several foods and medicinal products with an estimated daily exposure dose of 34 mg/kg. As a result of increased ENDS use, daily exposure to 1,2-PDO is rising. High levels of 1,2 PDO cause metabolic acidosis, hemolysis, neurotoxicity, and reduced renal clearing levels, especially in neonates who are more susceptible [65]. Toxic range is defined from 1 g/kg/day, and serious clinical symptoms occur at 3 g/kg/day [70]. Further 1,2-PDO was suggested to have, similar as ethanol, “alcohol-like” effects on the human central nervous system [71]. Massarsky et al. evaluated the neurobehavioral impact of 1,2-PDO toxicity in a zebrafish model, assessing two different concentrations (0.625% and 1.25%). In this study, 1,2-PDO was associated to a hyperactive swimming pattern, concluding that these substances may cause long-term neurobehavioral consequences [72].

## 4. Limitations

The present study corresponds to a narrative review with a systematic search criteria. We included publications with a more relevant scientific methodology, whose conclusions were more robust in order to clarify the different aspects related to NRT and ENDS. The limitation of a systematic review, given the novel nature of this research line, might be the risk of generating a confusing plot line given the heterogeneity of the methodology and conclusions obtained in the current literature. Therefore, we selected the most relevant articles to establish a clear plot line. Given the novel nature of research on ENDS, the volume of existing literature is heterogeneous and much lower than that on NRTs, whose studies have been developed since the last century. For ethical reasons, RCTs performed in humans comparing tobacco smoking to NRT and/or ENDS in pregnant women can never be done. Moreover, the majority of the included studies are animal studies, with their known limitation when extrapolating conclusions to humans.

## 5. Conclusions

Conventional tobacco smoking during pregnancy is still a major public health concern. Recent studies on NRT and ENDS do not support their use during gestation. NRT during pregnancy cannot be considered as a safe alternative to conventional tobacco smoking, but might be considered as “less harmful”. ENDS, even if “nicotine free”, may contain potentially toxic compounds and, therefore, cannot be considered harmless. Obviously, the best strategy during pregnancy is not to use any kind of nicotine-releasing device. As a rational precautionary measure, the issue is not whether vaping or NRTs should be recommended, however, it should be tried in order to reduce the nicotine dose to the minimum. In some cases, this may be vaping with a flavor-only product because, at least, limited research shows that such vaping can relieve the urge to smoke in about 30% of smokers [73]. However, under a damage-reduction recommendation, perhaps it will be mandatory to design efficacy and security studies. The safety of NRT and ENDS is increasingly being called into question. Therefore, it is imperative to perform more studies in order to evaluate the collateral and harmful effects of these delivery strategies on human health, especially during pregnancy. Investigations on side-effects and toxicity derived from the use of alternative strategies such as NRT and ENDS are considered as a novel line of research. The available studies are of a preclinical type, nevertheless they show potential clinical implications. There is currently no clear evidence of impact on fetal brain development, but recent research suggests that the current guidelines should be reconsidered.

## Figures and Tables

**Table 1 ijerph-16-05113-t001:** Recommendations in current guidelines for nicotine replacement therapy (NRT) and ENDS use during pregnancy.

Organisation	NRT	ENDS
RACGP [43]	Consideration if smoking cessation was unsuccessful Informed consent about risks and benefits Oral NRT as first-line therapy	No recommendation available
NICE [44]	Consideration if smoking cessation was unsuccessful without medication Only prescribe once women stop smoking; only prescribe 2 weeks of NRT	No recommendation available
CAN-ADAPTT [45]	Limited evidence in pregnancy Benefits may outweigh potential risks Consider (if counselling was ineffective) oral NRT as first-line therapy	No recommendation available
USPSTF [39]	No recommendation due to lack of evidence	Not recommended due to lack of evidence
ACOG [10]	Limited evidence in pregnancy NRT use should be supervised and restricted to women with a clear intention to quit smoking	No recommendation due to lack of evidence on pregnant women and their fetuses
WHO [46]	Limited evidence in pregnancy	Smokers should first be encouraged to quit smoking and nicotine addiction by using a combination of already-approved treatments ENDS use poses serious threats to adolescents and fetuses

ACOG: American College of Obstetricians and Gynecologists. CAN-ADAPTTL: Canadian Action Network for the Advancement, Dissemination and Adoption of Practice- informed Tobacco Treatment. NICE: National Institute for Health and Care Excellence. RACGP: Royal Australian College of General Practitioners. USPSTF: United States Preventive Services Task Force. WHO: World Health Organization. NRT: nicotine replacement therapy. ENDS: electronic nicotine delivery systems.

**Table 2 ijerph-16-05113-t002:** In vitro, animal and human studies assessing NRT impact on pre/postnatal brain development.

Author (Year)	Aim of Study	Type of Study	Methods	Outcomes	Key Results
Thomas et al. [47]	To examine the behavioral effects of nicotine exposure in the rat during the third trimester equivalent of the human brain growth spurt.	Animal study	Sprague-Dawley rat pups were exposed to nicotine (6.0 mg/kg/day) from postnatal days (PD) 4–9 via an artificial rearing procedure. Two control groups were employed, an artificially reared control group and a normally reared control group. Activity level was measured on PD 18–19.	Women who use tobacco products during late gestation may place their fetuses at risk for hyperactivity later in life, particularly during early adolescence.	-Nicotine-exposed subjects were significantly overactive compared with both control groups, which did not differ significantly from one another. -This behavioral alteration was observed in the absence of nicotine-induced body weight deficits.
Mahar et al. [28]	To explore the consequences of chronic developmental nicotine exposure on cerebral neuroplasticity in the offspring. Authors focused on two forms of neural plasticity in the dentate gyrus (DG) of the hippocampus that are highly sensitive to the environment: granule cell neurogenesis and long-term potentiation (LTP).	Randomized trial in animal models Nicotine = 7 Sterile saline = 9	Pregnant rats were implanted with osmotic mini-pumps delivering either nicotine or saline solutions. Offspring were chronically exposed to nicotine in utero and then through breastmilk. Plasma nicotine and metabolite levels were measured in dams and offspring. Corticosterone levels, DG neurogenesis, and glutamatergic electrophysiological activity were measured in pups.	Synaptic plasticity of offspring is disrupted in utero and in breastfed dams passively exposed to nicotine in an NRT-like model. It does reveal changes affecting connectivity in the growing brain.	-Juvenile (P15) and adolescent (P41) offspring exposed to nicotine throughout prenatal and postnatal development displayed no significant alteration in DG neurogenesis compared with control offspring. -Chronic NRT-like exposure during prenatal and postnatal development does not alter basal stress hormone levels in pups. -NRT-like nicotine exposure significantly increased LTP in the DG of juvenile offspring as measured in vitro from hippocampal slices, suggesting that the mechanisms underlying nicotine-induced LTP enhancement previously described in adult rats are already functional in pups.
Roy et al. [48]	To evaluate cellular morphology and regional architecture in the juvenile and adolescent hippocampus and the somatosensory cortex in rats exposed to nicotine prenatally.	Animal study	Pregnant rats were given nicotine throughout gestation via minipump infusion of 2 mg/kg/day On postnatal days 21 and 30, brains were perfusion fixed, coronal slices were taken and the morphology of the dorsal hippocampus and somatosensory cortex was characterized.	These data demonstrate that prenatal nicotine exposure compromises neuronal maturation, leading to long-lasting alterations in the structure of key brain regions involved in cognition, learning, and memory.	-In the hippocampal CA3 region and dentate gyrus, a decrease in cell size was found with corresponding decrements in cell layer thickness, and increments in cell packing density. -Smaller, transient changes were seen in CA1. In layer five of the somatosensory cortex, although there was no significant decrement in the average cell size, there was a reduction in the proportion of medium-sized pyramidal neurons, and an increase in the proportion of smaller, nonpyramidal cells. All regions showed elevated numbers of glia.
Slotkin et al. [49]	To assess nicotine effects on fetal brain development.	Animal study	Nicotine was administered to rats throughout gestation or in adulthood (postnatal days PN90-107), using regimens that reproduce plasma levels in smokers, assessing effects on serotonin (5HT) receptors, the 5HT transporter, and responses mediated through adenylyl cyclase (AC).	Animal studies show that nicotine itself leads to deficits in the number of neurons, neuronal and synaptic damage, and cognitive dysfunction. The replacement of tobacco with NRT during pregnancy could not be safe due to experimental evidence concerning nicotine’s injurious and enduring effects on neuronal systems.	-Prenatal nicotine exposure elicited persistent suppression of 5HT1A receptors and upregulation of 5HT2 receptors, effects that were selective for males and that first emerged in young adulthood. -AC activity was reduced and there was uncoupling of receptor-mediated responses. -With nicotine exposure restricted to adulthood, there were few changes in 5HT synaptic proteins during treatment or in the first 2 weeks post-treatment, distinctly different from the robust alterations seen earlier with similar nicotine regimens given in adolescence.
Slotkin et al. [50]	To assess the effects of tobacco exposure on brain cells and lipid peroxidation in Rhesus monkeys.	Animal study	Rhesus monkeys were exposed to environmental tobacco smoke (ETS) during gestation and through 13 months postnatally, or postnatally only (6–13 months). At the conclusion of exposure, cerebrocortical regions and the midbrain for cell damage markers and lipid peroxidation were examined.	Perinatal or postnatal ETS exposure in primates elicits changes in brain cell development.	-For perinatal ETS, two patterns were seen in the various regions: (1) cell loss (reduced DNA concentration) and increases in cell size (increased protein/DNA ratio), (2) replacement of larger neuronal cells with smaller and more numerous glia (increased DNA concentration, decreased protein/ DNA ratio). -Perinatal ETS exposure reduced the level of lipid peroxidation as assessed by the concentration of thiobarbituric acid reactive species, whereas postnatal ETS did not.
Morales-Suarez-Varela et al. [51]	To examine whether maternal smoking and use of nicotine substitutes during the first 12 weeks of pregnancy increased the prevalence of congenital malformations.	Danish National Birth Cohort, prospective data. 20,603 were exposed to tobacco smoking during the first 12 weeks of pregnancy.	Birth outcomes were collected by linkage to the Central Population Register, the National Patients Register, and the National Birth Register. Congenital malformations from the Hospital Medical Birth Registry.	No increase in congenital malformations related to prenatal tobacco smoking. An increased risk of malformations in non-smokers using nicotine substitutes.	-Children exposed to prenatal tobacco smoking had no increase in congenital malformations prevalence in both crude and adjusted analyses. -Children born to nonsmokers, but who used NRT, had a slightly increased relative congenital malformations prevalence ratio; relative prevalence rate ratio was 1.61 (95% confidence interval 1.01–2.58), which represents a 60% increased risk. -When the analysis was restricted to musculoskeletal malformations, the relative prevalence rate ratio was 2.63 (95% confidence interval 1.53–4.52).
Strandberg-Larsen et al. [54]	To assess if the use of NRT during pregnancy increases the risk of stillbirth.	Danish National Birth Cohort, prospective data 87,032 singleton pregnancies.	Outcome of pregnancy was identified by register linkage, with <1% loss to follow-up. Cox regression analyses to estimate the hazard ratio (HR) and 95% CI of stillbirth according to the use of NRT, type of NRT use, and a combination of NRT use and smoking.	The use of NRT during pregnancy does not increase the risk of stillbirth.	-A total of 495 pregnancies (5.7 in 1000 births) ended in stillbirth, eight of which were among NRT users. -After adjustment for confounders, women who used NRT during pregnancy had an HR of 0.57 (95% CI 0.28–1.16) for stillbirth compared with those who did not use NRT -Smoking during pregnancy was associated with an increased risk of stillbirth (HR 1.46, 95% CI 1.17–1.82). -Women who both smoked and used NRT had a HR of 0.83 (95% CI 0.34–2.00) compared with nonsmoking women who did not use NRT
Cooper et al. [55]	To compare: (1) At delivery, the clinical effectiveness and cost- effectiveness for achieving biochemically-validated smoking cessation of NRT patches with placebo patches in pregnancy. (2) In infants at 2 years of age, the effects on behavior, development, and disability.	Randomized, placebo-controlled, parallel group 1050 pregnant smokers 521 NRT/529 placebo.	Participants were randomly assigned (1:1) to receive 8 week courses of NRT patches (15 mg/16 h) or matched placebo. Follow-up at 4 weeks after randomization, delivery, and until infants were 2 years old. Participants: self-reported, prolonged abstinence from smoking between a quit date and childbirth, validated at delivery by carbon monoxide (CO) measurement and/or salivary cotinine (COT). Infants, at 2 years: absence of disability or problems with behavior and development. Economic: cost per quitter.	NRT patches had no enduring significant effect on smoking in pregnancy; however, 2 year olds born to women who used NRT were more likely to have survived without any developmental impairment.	-Numbers of adverse pregnancy and birth outcomes were similar in both trial groups, except for a greater number of caesarean deliveries in the NRT group. -At 1 month after randomization, the validated cessation rate was higher in the NRT group (21.3% vs. 11.7%, OR, (95% CI), 2.05 (1.46 to 2.88)). -At delivery, no difference between groups’ smoking cessation rates: 9.4% in the NRT and 7.6% in the placebo group (OR (95% CI), 1.26 (0.82 to 1.96)). -Infants: at 2 years, analyses were based on data from 888 out of 1010 (87.9%) singleton infants (including four postnatal infant deaths) 445/503 (88.5%) NRT, 443/507 (87.4%) placebo, and used multiple imputation. In the NRT group, 72.6% (323/445) had no impairment compared with 65.5% (290/443) in placebo (OR 1.40, 95% CI 1.05 to 1.86).

**Table 3 ijerph-16-05113-t003:** In vitro and animal studies assessing ENDS impact on pre/postnatal brain development.

Author (Year)	Aim of Study	Type of Study	Methods	Outcomes	Key Results
Bahl et al. [64]	Compare sensitivity of human embryonic stem cells (hESC) and mouse neural stem cells (mNSC) to (adult) human pulmonary fibroblasts (hFP) after ENDS refill liquid exposure.	In vitro	Cell exposure to 35 refill liquids (*n* = 35) using MTT assay for cell metabolic activity assessment.	Embryonic stem cells (hESC, mNSC) are more sensitive than adult lung fibroblasts (hFP). Cytotoxic effects may cause embryonic loss or developmental defects during pregnancy.	15 refill samples showed moderate cytotoxity (IC 50: 0.1%–1%) to hESC and mNSC. 10 refill samples had little or no effect on hPF (IC50 > 1%).
Zahedi et al. [65]	Assessment of ENDS nicotine-containing refill fluids and their aerosols on neural stem cell (NSC) mitochondria.	In vitro	24 h cell incubation and exposure to refill fluids and aerosols compared with untreated controls.	ENDS refill fluids and aerosols provoke stress-induced mitochondrial hyperfusion (SIMH) in NSC. SIMH was accompanied by alterations in mitochondrial morphology and dynamics. ENDS are not as harmless as often claimed. Even short-term exposure can stress cells and lead to morbidity or disease.	SIMH increased at 0.3% nicotine concentration, and cell swelling increased at 0.5% and 1% nicotine concentration.
Lauterstein et al. [66]	Transcriptome RNA-sequencing of frontal brain cortex (FBC) in mice exposed to ENDS aerosols (±nicotine) compared with air-exposed controls.	Animal study (pregnant C57BL/6 mice)	Pre- and postnatal exposure via whole body inhalation. Analyses of gene expression in FBC.	ENDS exposure alters brain development, causing chronic neuropathology. Decrease in memory, cognition, and neurotransmission. Increase in hyperactive behavior, emotional behavior, and death. ENDS non-nicotine: significant gene expression changes in FBC. Components other than nicotine may affect neurodevelopment. ENDS are associated to adverse neurobiological and neurobehavioral outcome similar to early life conventional cigarettes.	Gene expression changes (GEC) Female: -ENDS non-nicotine: 2630 -ENDS nicotine: 1393 Male: -ENDS non-nicotine: 2615 -ENDS nicotine: 152
Smith et al. [67]	ENDS nicotine exposure during rapid brain growth period is associated with behavioral changes in adult mice.	Animal study (pregnant C57BL/6 mice)	Pre- and postnatal exposure to 2.4% nicotine in 1,2-PDO versus 0% nicotine/1,2-PDO. Assessment of cotinine levels and behavioral testing at 14 weeks of age. (Gestational day 15–19 and postnatal day 2–16 are equivalent to third trimester brain growth in humans.)	ENDS nicotine increases cotinine levels, activity, and number of head dipping and rearing, and increased cognitive flexibility. ENDS may cause persistent behavioral changes when exposure occurs during a period of rapid brain growth.	Cotinine levels: 2.4%/PG: 23.7 ± 4.2 ng/mL. 0%/PG: 2.8 ± 0.3 ng/mL.
Nguyen et al. [68]	Maternal ENDS aerosol exposure on murine offspring and assessment of impact on behavior and global DNA methylation in brain tissue.	Balb/C female mice	3 groups (*n* = 24). ENDS + nicotine (*n* = 8), ENDS nicotine-free (*n* = 8), and air (*n* = 8). Exposure 6 weeks before pregnancy, during pregnancy, and lactation. Behavioral assessment at 12 weeks of age. Epigenetic testing of brain tissues at day 1 and 20 days, and 13 weeks after birth.	Maternal ENDS + nicotine aerosol causes short-term memory deficits, reduced anxiety, and hyperactivity in offspring. Cognitive and epigenetic changes were observed in the offspring. The use of ENDS during pregnancy may have hitherto undetected neurological consequences on newborns.	Epigenetic testing: ENDS+nicotine-free aerosol showed statistically significant higher global DNA methylation compared with the air group at day 1 and 20. At week 13, no significant global DNA methylation change was observed in hippocampus. Significant alterations of 13 key genes were detected.

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
