# Peer review of "Impact of Nicotine Replacement and Electronic Nicotine Delivery Systems on Fetal Brain Development"

_ijerph, 2019, doi:10.3390/ijerph16245113_

Round 1

Reviewer 1 Report

This manuscript is a review of the state of knowledge regarding nicotine effects on fetal and subsequent brain development with a particular focus on comparing NRT and ENDS to cigarette smoking. The explosion of vaping makes this an interesting and worthwhile topic. In general, the manuscript does not seem sufficiently connected to meaningful clinical outcomes; additionally the lack of available data, particularly for ENDS (and the fact that available data may not reflect current product usage) limits enthusiasm.

Comments:

The abstract could include more information about the authors' conclusions.   Line 38, it would be useful to give an example or two of the postnatal impacts. If there are any estimates in the literature of the overall impacts in the literature (eg, number with nicotine-related behavioral problems) that would also be informative - eg, annual impacted births.  This may not exist though. Lines 42-45 describe NRT as having been introduced in the past decade and suggest that their efficacy and safety are uncertain, but I'm not sure that any of these statements is accurate as written - the source the authors cite here is itself 11 years old. It may be that the authors are referring to these being newly tried in pregnant smokers, but if so that should be explicit, and if that's the case studies go back some way - see for example Benowitz 1991, JAMA, 266, 3174-7. Lines 50-51: suggest citing source(s) for statements that ecigs are used for cessation but effectiveness and safety are unclear.   Line 81-83: what are the implications of these metabolic differences? It would help the reader if the review included more division into sections with headers. In the introduction there should be some discussion of the different ENDS generations and their relevant differences.  The various national guidelines referenced in 2.2 need citations. Are there more recent studies that have examined NRT vs smoking vs neither to follow up on the 2006 Morales et al study?  If so this should be included given the age of the latter. Lines 190-92: the authors should spell out what they mean here in terms of adverse second generation effects. The conclusions seem very cursory. What additional research is needed to better answer the limitations the authors point out? If the main limitation is lack of RCTs, that seems unlikely to be resolved.

Reviewer 2 Report

This very well done review would benefit from some modest re-framing and a bit more care in word selection. Suggested modifications are as follows:

1.  In paragraph 1 of the intro, four studies are referenced (1-4) in terms of nicotine exposure during pregnancy. These studies document tobacco smoke exposure during pregnancy.  In the context of this paper, this is a major error, since the implied purpose of the paper should be to differentiate the risks posed by smoking, vaping or use of NRTs during pregnancy. This statement, and others in the paper (if any) should clearly differentiate exposures to tobacco smoke from exposure just to nicotine, as in vaping or NRTs. 

2. Typo, iine 7 paragraph 1, "consume" should be "consumption"

3.  Also, in paragraph 1, toward the end, with reference to (9) -- this should be restated as ENDs use has been associated with 7-fold higher use of conventional cigarettes among minors, but opinions vary as to whether this finding implies causation or an enhanced willingness to experiment with prohibited substances by some teens.  I mention this because this particular NAS conclusion has been highly criticised. 

4.  Since no one would ever recommend use vaping or use of NRTs in pregnant women who are not smokers, the question is not absolute hazard, but relative risk among pregnant women unable or unwilling to quit. We know enough about nicotine exposure during pregnancy in animal models to know that it seems very likely that such exposure poses significant risk to the fetus in pregnancy.  We also know that exposure to cigarette smoke poses substantial risk to pregnant women.  What we do not know is whether this difference in risk is substantial enough to recommend use of vaping or NRTs in pregnant women unable or unwilling to discontinue use of nicotine in pregnancy.

5.  For ethical reasons, a randomised trial comparing smoking to vaping to NRT use in pregnant women can never be done.

6. It would be of value to have a randomized animal study comparing smoking to vaping to NRT use would be of value in this regard, but no such study has been done to date.

7.  As a rational precautionary measure, the issue is not whether vaping or NRTs should  be recommended, but that the woman be urged to use the lowest dose of nicotine needed to satisfy her urge to smoke. In some cases, this may be vaping with a flavor-only product because of at least limited research showing that such vaping can relieve the urge to smoke in about 30% of smokers. https://DOI.org/10.1136/tc.2009.033498  

--- the ideas in items 4-7 above should be reflected in the introduction, discussion and conclusions, and possibly in the Abstract

Reviewer 3 Report

Thank you to the authors for their hard work and tremendous attention to detail.  I found the research presented here to be recent and relevant to the research community and IJERPH readership, more specifically.  I do believe this review is in sync with the scope of the special issue for which it was submitted.  Despite my enthusiasm for the current manuscript, I do have a few concerns that should probably be addressed in subsequent revisions. 

(1) Most importantly, no information was provided on the sequence of steps used to conduct the review.  Readers will not know or be able to interpret the scope of the search, if relevant databases were overlooked, or the specific criteria used to conduct the search.  In short, this research would not be replicable by others, which is a fundamental requirement of scientific research today.  Care should be taken to include a Methods section that outlines the steps they used in their search and their review. 

(2) Literature cited on NRT appears to heavily outweigh that cited for ENDS.  While this is to be expected given the newness of the research with ENDS, this should be acknowledged and care taken to make certain all of the research on ENDS is included (hence, the need for a well-defined Procedures section within the Methods section). 

(3) While this is not a 'systematic review of the literature' in a formal sense, reviews today should, in my opinion, include a fairly comprehensive interpretation of the of the literature cited (direction of findings, effect sizes, sample sizes, sample types, etc).  As currently written, content is reviewed, tabularized, and a brief conclusion provided.  For a review of research to be valued by others, it must be comprehensive in scope (at least within its own predefined boundaries)--and insightful in its detail.  Future drafts of this paper should include a more insightful, detailed set of findings to make the fruits of their review truly accessible and useful to others working in this important area of public health.

In conclusion, I am grateful to the authors for their hard work and willingness to pursue this line of very important research. In its current form, however, the paper may be too early in development. With the above modifications and those of the others reviewers, this review could be great interest and value to the readership of IJERPH.  Thank you for sharing it with me for review. 

Round 2

Reviewer 1 Report

Thank you for the opportunity to review this revision. I appreciate that the authors seem to have sought to address the comments, though I'm not sure how successfully they did so. I also have questions about whether there are sufficient data in the literature to enable an informative review. 

The authors continue to refer to NRT as new (now lines 50-52) which is confusing. ENDS may have been originally designed to assist with cessation but I don't know that I would claim the same about the products currently in use. It is odd to state that both NRT and ENDS are new products designed to aid cessation. Lines 69-70: ecigs are sometimes used as a cessation tool but that is certainly not the only motivation and it's not clear that it's even the primary one. Among younger population it appears not to be. I appreciate the new clarity about why the impact of pregnancy on nicotine metabolism is important. I'm not sure that there's an issue with clinical recommendations, but certainly the standard NRT dose may be insufficient to ameliorate withdrawal as a result of the metabolic impact. Do not agree that pregnant women should be encouraged to use the lowest dose of nicotine that will satisfy urges. I get the point but would revise that statement. Satisfying may require a pretty high dose.

Author Response

see word document

Reviewer 3 Report

Thanks to the authors for responding to my concerns and suggestions.  While the edits were indeed responsive, there are some grammatical issues that need to be resolved. Please have an editor well-versed in English thoroughly read the next draft of the manuscript prior to resubmission (e.g., pg 1, ln 25; pg 2, ln 83).  

Author Response

see word document
